# Present-day and future Greenland Ice Sheet precipitation frequency from CloudSat observations and the Community Earth System Model

Jan T. M. Lenaerts[1,*], M. Drew Camron[1,*], Christopher R. Wyburn-Powell[1], and Jennifer E. Kay[1,2]

[1]Department of Atmospheric and Oceanic Sciences, University of Colorado Boulder, Boulder, CO 80309
[2]CIRES, University of Colorado Boulder, Boulder, CO 80309
[*]These authors have contributed equally to this work.

**Correspondence:** Jan Lenaerts (Jan.Lenaerts@colorado.edu)

**Abstract.** The dominant mass input component of the Greenland Ice Sheet (GrIS) is precipitation, whose amounts and phase are poorly constrained by observations. Here we use spaceborne radar observations from CloudSat to map the precipitation frequency and phase on the GrIS, and use those observations, in combination with a satellite simulator to enable direct comparison between observations and model, to evaluate present-day precipitation frequency in the Community Earth System Model (CESM). The observations show that substantial variability of snowfall frequency over the GrIS exists, that snowfall occurs throughout the year, and snowfall frequency peaks in Spring and Fall. Rainfall is rare over the GrIS, and only occurs in regions under 2000 m elevation and to the peak summer season. Although CESM overestimates the rainfall frequency, it reproduces the spatial and seasonal variability of precipitation frequency reasonably well. Driven by a high-emission, worst-case Representative Concentration Pathway (RCP) 8.5 scenario, CESM indicates that rainfall frequency will increase considerably across the GrIS, and will occur at higher elevations, potentially exposing a much larger GrIS area to rain and associated meltwater refreezing, firn warming, and reduced storage capacity. This technique can be applied to evaluate precipitation frequency in other climate models, and can aid in planning future satellite campaigns.

## 1 Introduction

The Greenland Ice Sheet (GrIS) contains the largest volume of ice on the Northern Hemisphere, equivalent to 7.4 meter sea level equivalent (Morlighem et al., 2017). While the GrIS has been losing mass since the 1970s, and likely also in earlier episodes in the 20th century (Kjeldsen et al., 2015; Kjær et al., 2012), observations indicate that GrIS mass loss has accelerated since the mid-1990s to early 2000s (Van den Broeke et al., 2016; Mouginot et al., 2019).

Mass loss is equivalent to a negative ice sheet mass balance (MB). Negative MB, in turn, implies that the ice sheet surface mass balance (SMB) is lower than the total flux of ice across the GrIS grounding line (D, (Lenaerts et al., 2019)). The aforementioned enhanced GrIS mass loss has been predominantly driven by a progressively declining SMB (Shepherd et al., 2020). In contrast, D has remained relatively constant since the 2000s, despite interannual (Enderlin et al., 2014) and seasonal variations (King et al., 2018).

GrIS SMB is predominantly governed by the difference between precipitation (snowfall and rainfall) and meltwater runoff from snow and ice melt, with surface and blowing snow sublimation an order of magnitude lower (Lenaerts et al., 2012). Recent GrIS SMB decrease is driven by enhanced surface melting and runoff (Van den Broeke et al., 2016; Trusel et al., 2018), caused by increasing atmospheric temperatures (Van Angelen et al., 2014; Fettweis et al., 2012) and persistent anomalously high large-scale atmospheric blocking over the GrIS (Fettweis et al., 2013; Belleflamme et al., 2015). In contrast, climate modeling and airborne radar observations indicate that GrIS precipitation has remained relatively constant (Van den Broeke et al., 2016; Montgomery et al., 2020; Fettweis et al., 2020), with only an increase over parts of the interior (Csatho et al., 2014; Lewis et al., 2019). Throughout the remainder of the 21st century, sustained atmospheric warming is expected to cause continued GrIS mass loss (Pattyn et al., 2018), but the potential role of increasing precipitation on mitigating that GrIS mass loss is highly uncertain.

The amount of precipitation that falls on the GrIS ablation zone (areas where local SMB $< 0$) during winter determines the depth of the snow layer that is melted away, and thereby, controls the timing of bare, low-albedo ice exposure in summer (Noël et al., 2019; Ryan et al., 2019). For example, the 2017-2018 winter was a very wet winter on Greenland, delaying the ice exposure onset to late in the summer season, which led to anomalously low melt and runoff, and high SMB, in 2018. Additionally, snowfall events during the melt season can significantly limit subsequent melting (Noël et al., 2015). In the GrIS accumulation zone (where local SMB $> 0$), the depth of the winter snowpack controls the availability of 'cold content', i.e. energy to refreeze and locally store surface meltwater that percolates into it, thereby preventing runoff of that water into the ocean. In regions of very high snow accumulation, such as Southeast Greenland, the winter snowpack can also act to thermally insulate warm firn containing liquid water below (the firn aquifer, Forster et al. (2014)) from the cold atmosphere aloft.

Despite its importance for the GrIS mass balance and firn processes, it is notoriously challenging to retrieve direct observations of precipitation on the GrIS. Precipitation gauges struggle from undercatching snowfall, and those gauges that overcome these issues are large and expensive, and hence difficult to deploy in the field. Alternatively, snow accumulation rates can be derived from firn cores, ground-based and airborne snow radar, and cosmic ray counters, but these observations do not allow to separate precipitation from other surface mass balance processes such as sublimation, vapor deposition, and blowing snow redistribution (Lenaerts et al., 2019). Moreover, precipitation amounts vary greatly across the GrIS and across seasons and years, indicating the need of distributed and long-term observations to capture the full extent of GrIS precipitation. The phase of the precipitation on the GrIS is even more uncertain, as precipitation phase is determined by complex thermodynamic and cloud micro-physical processes that are poorly constrained over much of the Polar Regions.

Over the last decade, new satellite remote sensing technology has enabled the direct observation of precipitation in Polar Regions, including over the GrIS. Specifically, the CloudSat satellite has an active cloud-profiling 94-GHz radar and has been observing polar clouds and precipitation since 2006 (Stephens et al., 2002). Unlike precipitation radars such as TRMM and GPM, which are designed to target heavy tropical precipitation, CloudSat is sensitive to the light precipitation and snow that falls at high latitudes. Due to its orbit and its sensitivity, CloudSat is currently the only radar in space that measures precipitation at high latitudes. CloudSat observations have been used to assess Antarctic Ice Sheet precipitation rates (Palerme et al., 2014, 2016; Boening et al., 2012; Milani et al., 2018; Lemonnier et al., 2020) and GrIS precipitation rates (Bennartz et al., 2019).

While CloudSat observations are unique in their measurement of polar precipitation, they are not without limitations. In particular, CloudSat radar reflectivity profiles are contaminated by ground clutter in the bottom kilometer of the atmosphere, which limits their ability to assess surface precipitation. In addition, converting CloudSat reflectivity observations into precipitation amount requires assumptions about the drop size distribution and shape. To circumvent these limitations, many precipitation studies have applied thresholds to CloudSat's near-surface radar reflectivity to estimate near-surface precipitation frequency (Haynes et al., 2009; Ellis et al., 2009; Smalley and L'Ecuyer, 2015). Recently, these near-surface radar reflectivity derived precipitation frequencies have been compared to climate model output in a scale-aware and definition-aware framework (Kay et al., 2018). Here, we use this framework to compare present-day GrIS precipitation frequency between observations (Cloud-Sat) and an Earth System Model (CESM). After understanding present-day biases, we assess future 21st century changes in precipitation frequency over the GrIS, and discuss the implications for future radar missions. We start this paper with a presentation of our framework for comparing models and observations and a description of the model simulations (Section 2), followed by results (Section 3). Section 4 presents a discussion and conclusions.

## 2 Data and methods

### 2.1 Scale-aware and definition-aware framework for evaluating simulated precipitation frequency

Evaluating precipitation simulated by Earth System Models with satellite observations is challenged by the scale differences (model grids are ∼100 km, while CloudSat footprints are ∼1 km), and because of inherent differences in the definition of precipitation between models and observations. In addition, CloudSat suffers from ground clutter, which leads to, for example, missing up to 25% of the light snow producing mixed-phase clouds over central Greenland (Bennartz et al., 2019; McIlhattan et al., 2019). To address these challenges, the science community has developed a software package called Cloud Feedbacks Model Intercomparison Project (CFMIP) Observational Simulator Package (COSP, Bodas-Salcedo et al. (2011)). COSP contains a sub-column generator and instrument forward models, called simulators, to convert raw model output at the model grid scale into pseudo-satellite observations at the satellite footprint. As such, COSP outputs can be directly compared to equivalent satellite observations in a scale-aware and definition-aware framework. For this study, we use the Quickbeam radar simulator (Haynes et al., 2007) to simulate modelled CloudSat reflectivity profiles. Subsequently, following Kay et al. (2018), we calculate near-surface precipitation frequency based on thresholding the modeled near-surface CloudSat reflectivity. Using this framework, we are able to directly compare modelled and observed CloudSat near-surface precipitation frequency. The observations we use are gridded observations of 2C-PRECIPITATION-COLUMN (2CPC hereafter) CloudSat near-surface precipitation frequency (Ellis et al., 2009) during 11 years (June 2006 - May 2016). This grid has a 1 x 1 degree horizontal resolution that aggregates all CloudSat 2CPC observations. Since CloudSat has been operating on daytime only mode since 2011, which might potentially introduce biases that are not considered in this study. The model and the observations use the same reflectivity thresholds (Kay et al., 2018) for assessing near-surface precipitation frequency. Here we use the 'light snow' (near-surface (960–1,440 m above the surface) attenuated radar reflectivity (dBZ) between -15 and -5 and near-surface air temperature (T) < 273 K) , 'snow' (dBZ > -5 and T < 273 K), 'light rain' (-15 < dBZ < -5 and T > 275 K), and 'rain' (dBZ

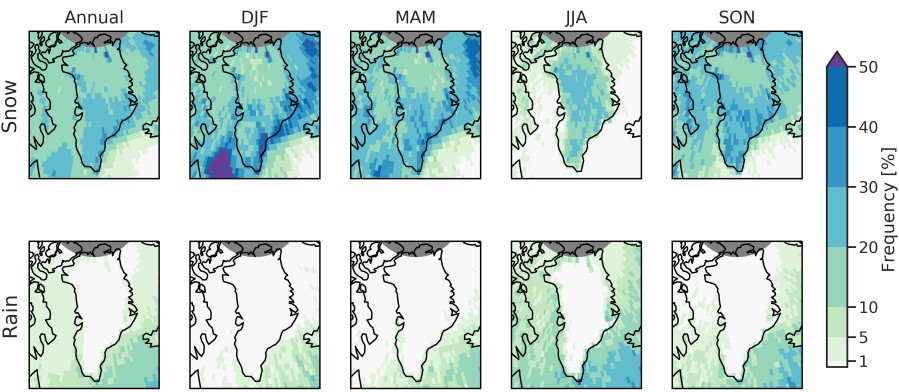

**Figure 1.** Annual (left) and seasonal (DJF, MAM, JJA, SON, from left to right) mean total snow (light snow + snow, top) and total rain (light rain + rain, bottom) frequency derived from CloudSat cloud-profiling radar (2CPC: 2006–2016).

> -5 or heavily attenuated, and T > 275 K) categories, as heavy precipitation (as defined by Kay et al. (2018)) does not occur on the GrIS. We define precipitation frequency as the ratio between the number of time steps with precipitation and the total number of time steps. If averaged across an area, such as the ice sheet or elevation bin, frequency is defined as the average frequency of all grid cells contained within that area. For more details regarding the methodology, refer to Kay et al. (2018).

## 2.2 Model simulations with CloudSat near-surface precipitation frequency diagnostics

We assess GrIS precipitation simulated by the Community Earth System Model (CESM) version 1 with the Community Atmosphere Model version 5 (CESM1-CAM5, CESM hereafter, Hurrell et al. (2013)). CloudSat near-surface precipitation frequency diagnostics were implemented in COSP version 1.4 (Kay et al., 2016b, 2018). While this study uses COSP1.4, the CloudSat-based diagnostics described here are also available for the broader scientific community within the latest COSP version, COSP 2 (Swales et al., 2018).

In order to evaluate present-day GrIS precipitation and to assess GrIS precipitation in a warmer future world, we ran CESM using the worst-case Representative Concentration Pathway (RCP) 8.5 greenhouse gas emissions scenario. The simulations span 90 years (2006 to 2095) and was initialized in 2006 from member 1 of the CESM1 Large Ensemble (Kay et al., 2015). The same simulation has been used to assess the influence of global warming on rising cloud heights (Takahashi et al., 2019).

## 3 Results

### 3.1 Present-day precipitation from CloudSat

First we explore the present-day spatial and temporal precipitation frequency patterns that have been observed by CloudSat from 2006 to 2016. Figure 1 shows the observed annual and seasonal mean spatial patterns of total snowfall (i.e. the sum of 'light snow' and 'snow') and rainfall (the sum of 'light rain' and 'rain') frequencies on the GrIS. The annual mean snowfall

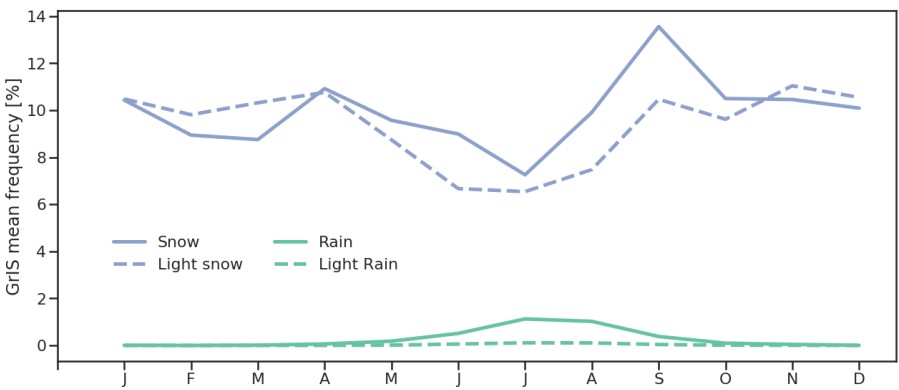

**Figure 2.** Seasonal cycle (January to December) of Greenland Ice Sheet averaged snowfall, light snowfall (blue: solid, dashed) and rainfall, light rainfall (green: solid, dashed) frequency obtained by CloudSat cloud-profiling radar observations (2CPC: 2006–2016).

on the GrIS varies from ∼10% in the dry, high-elevation northern GrIS, to >30% over Southeast Greenland. The interior experiences snowfall most frequently in the summer (JJA, >20%), whereas most snow in the coastal regions falls in winter (DJF), and to a less extent in Spring (MAM) and Fall (SON). Observed snowfall frequency over the oceans surrounding the GrIS is highest in the winter, particularly in the Labrador Sea (southwest of the GrIS), where winter snowfall frequency exceeds

50%. In summer, snowfall does not occur over the oceans around the GrIS. Rainfall over the interior of the GrIS is negligible throughout the entire year. Rain occurs in summer, albeit rarely ($< 10\%$),over the marginal, low-elevation zones of the GrIS. Summer rainfall frequency is largest over the North Atlantic ocean, southeast of the GrIS (>30%). Averaged across the GrIS, light snow and snow show similar seasonal cycles (Figure 2), and vary from 6% in summer to ∼10% over the rest of year, with peaks in spring and fall. Light rain does not occur on the GrIS, and rain only occurs from June to September, with maximum

values (∼2%) in July and August.

A unique perspective on the CloudSat precipitation frequency climatology across the GrIS can be offered by analyzing their gradients with respect to surface elevation (Figure 3). Snow frequency varies moderately with elevation, and the highest snow frequencies (>20%) are found at the lowest elevations ($< 200$ m above sea level (a.s.l.)) as well as between 1500 and 2000 m a.s.l.. The latter maximum can be explained by the strong topographically forced snowfall in Southeast GrIS, where the

maximum snowfall occurs at these elevations. When classifying the snow frequency, heavier snow peaks at these elevations and otherwise decreases with elevation, while light snow frequency clearly increases with height and dominates heavy snow above 2000 m a.s.l. (not shown). Rainfall frequency (which is dominated by rain, as light rain is almost zero everywhere (not shown)) does not exceed 2% anywhere on the GrIS, and rain is never observed above 2000 m a.s.l.. Note that, due to the hyperbolic shape of the GrIS and steep surface slopes along the margins, low-elevation areas occupy a very small fraction of

the ice sheet, while higher-elevation areas occupy a much larger fraction. This implies that, although all areas below 2000 m a.s.l. experience rain, all these elevation bands combined only occupy ≈ 38% of the ice sheet area.

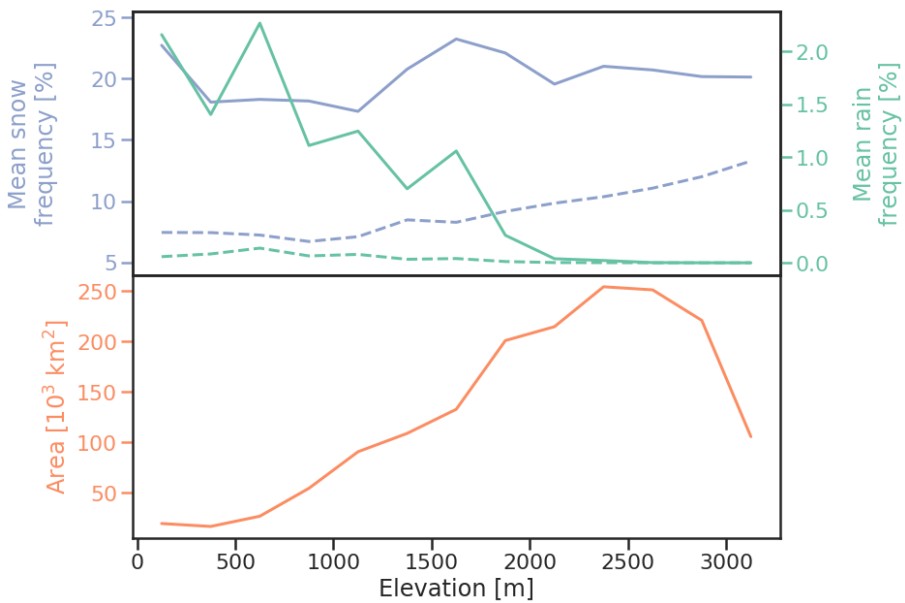

**Figure 3.** Top: CloudSat 2CPC (2006–2016 average) Greenland Ice Sheet snow, light snow (blue: solid, dashed) and rain, light rain (green: solid, dashed) frequency in 250-m elevation bins based on Greenland Ice sheet Mapping Project (GIMP, Howat et al. (2014)) topography. Bottom: total ice sheet grid-cell surface area in each of these elevation bins according to GIMP.

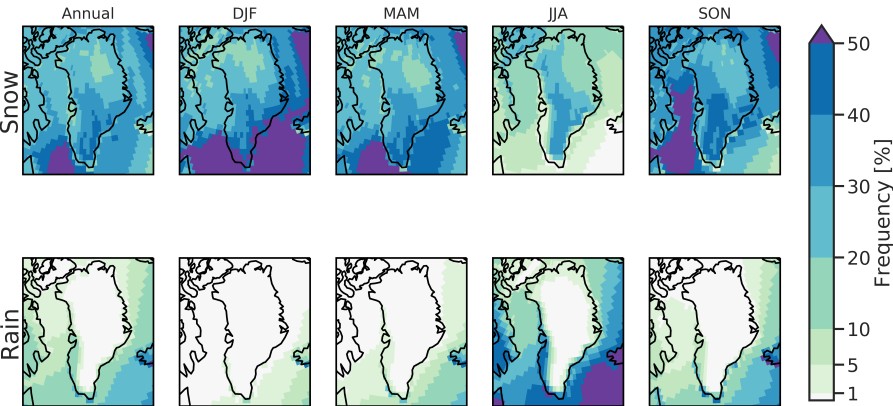

**Figure 4.** Present-day, annual (left) and seasonal (DJF, MAM, JJA, SON, from left to right) mean total snowfall (light snow + snow, top) and total rainfall (light rain + rain, bottom) frequency as simulated by CESM (2006–2020).

## 3.2 Present-day precipitation from CESM

We first present the CESM precipitation frequencies (Figure 4), and then compare them directly to CloudSat (Figure 5). The highest snowfall frequencies produced by CESM are found in the oceanic regions neighbouring the GrIS, the North Atlantic

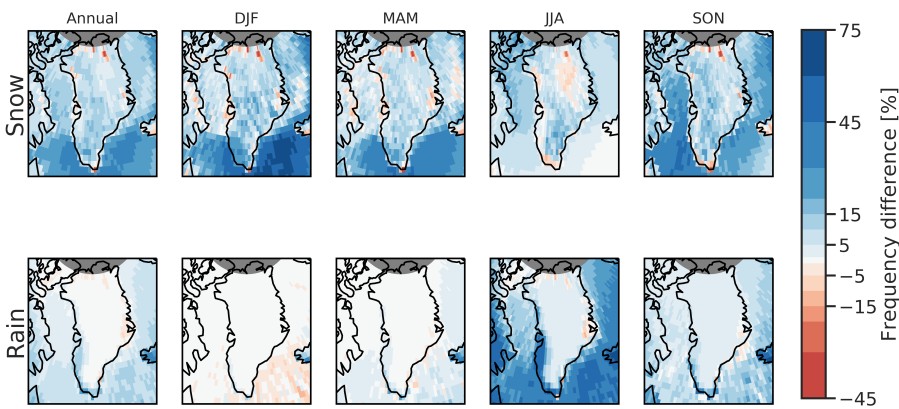

**Figure 5.** Present-day, annual (left) and seasonal (DJF, MAM, JJA, SON, from left to right) mean snowfall (top) and rainfall (bottom) frequency difference between CESM (2006–2020) and CloudSat 2CPC (2006–2016). Positive values indicate that CESM overestimates precipitation frequency relative to CloudSat.

and Baffin Bay along the southwest GrIS coast, in concert with what CloudSat shows. On the ice sheet, snowfall frequency is highest in the south (>40%), and decreases northward to low values of <20% in the high-elevation interior. CESM simulates a clear seasonal cycle in snowfall frequency, with highest frequency in winter and lowest in summer. CESM produces rainfall on the oceans around Greenland during most of the year, while GrIS rainfall is constrained to the summer season, and limited

to the coastal regions.

Next, we compare the CESM simulated precipitation frequency on the GrIS to the frequencies derived by CloudSat (Figure 5). Snowfall frequency over the GrIS is generally overestimated by CESM, especially in winter and fall (>15%). Over the surrounding oceans, CESM clearly produces more frequent snowfall than CloudSat, with up to 75% more frequent snowfall in the North Atlantic in winter. In contrast, interior GrIS summer snowfall frequency is slightly lower in CESM than in CloudSat.

In contrast with CloudSat, CESM only produces rainfall in the low-elevation GrIS coastal zones, and in summer, but the rain frequencies are clearly overestimated, especially over the western GrIS ablation zone and the oceans. CESM produces slightly lower rain frequencies in the North Atlantic compared to CloudSat in winter.

The seasonal cycle of precipitation frequency averaged over the GrIS, as shown in Figure 6, highlights seasonal variations in light snow and light rain frequencies as simulated by CESM. In summer, the only season in which light rain occurs according

to CESM, the simulated light snow frequency is smaller than in the other seasons. Throughout most of the year, the simulated light snow contributes more to the total snowfall frequency than the heavier snow. This heavier snow also exhibits less of a seasonal variability than the light snow. Similarly, light rain dominates the total rainfall across the Greenland ice sheet, as heavier rain does not occur.

Analyzing the differences between CESM and CloudSat with respect to elevation across the GrIS (Figure 7), we see that

CESM overestimates snowfall frequencies with 5 to 10% at all elevations. CESM also produces an increase of snow frequency with elevations from the coast to 2000 m a.s.l., which is not confirmed by CloudSat. With regards to rain, CESM clearly

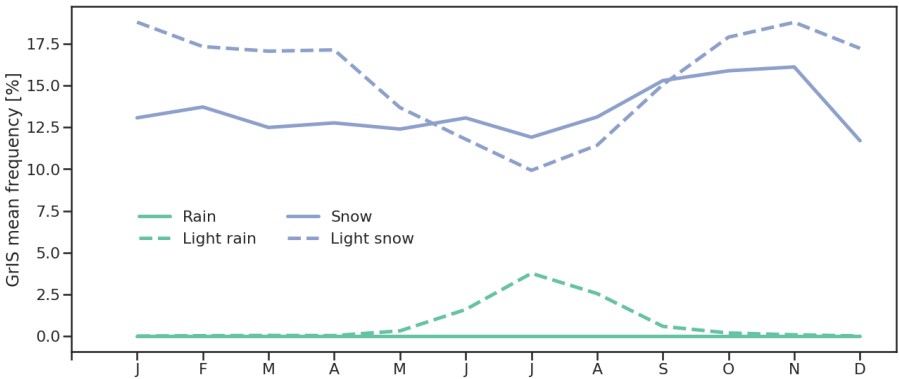

**Figure 6.** Present-day, Greenland Ice Sheet averaged snowfall, light snowfall (blue: solid, dashed) and rainfall, light rainfall (green: solid, dashed) frequency as simulated by CESM (2006–2020).

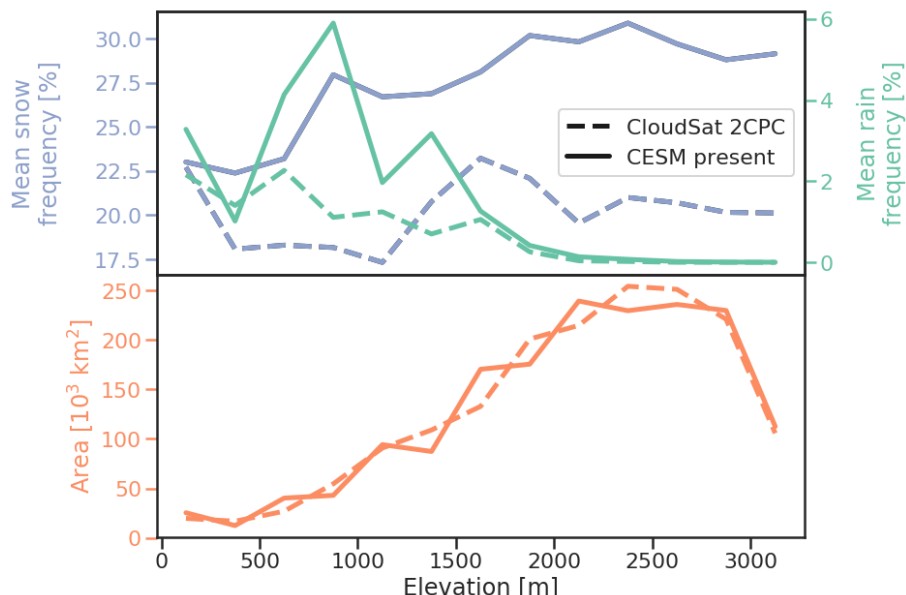

**Figure 7.** Top: Present-day, Greenland Ice Sheet averaged total snow (blue) and total rain (green) frequency in 250-m elevation bins based on GIMP topography, according to CloudSat 2CPC (dashed, 2006–2015) and CESM (solid, 2006–2020). Bottom: total ice sheet grid-cell surface area in each of these elevation bins according to CloudSat (dashed) and CESM (solid).

produces too high frequencies at lower elevations (double to triple the CloudSat frequency). On the other hand, the model correctly simulates the clear decrease in rain frequency above 1500 m a.s.l., and agrees with CloudSat in that it simulates no rain above 2000 m a.s.l..

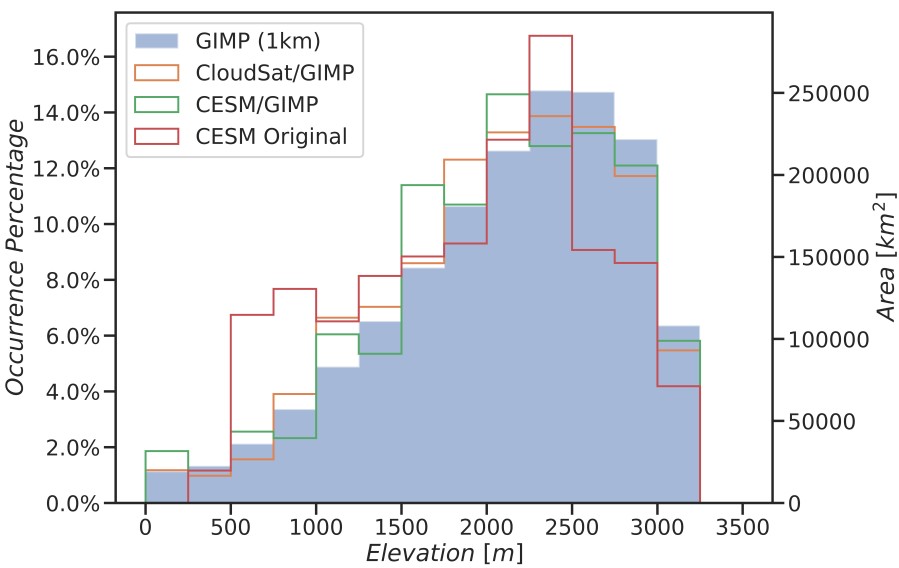

**Figure 8.** Comparison of GrIS hypsometry in GIMP (grey), CloudSat regridded to GIMP (orange), CESM regridded to GIMP (green), and original CESM (red).

A part of these discrepancies between CESM and CloudSat may be ascribed to CESM (at its horizontal resolution of 1 degree) not resolving the steep topography and related surface climate and precipitation gradients of the marginal GrIS. This is illustrated in Figure 8, which shows that the original CESM grid overestimates the extent of low-elevation areas and underestimates the extent of high-elevation areas of the GrIS. While we have attempted to correct for this by regridding the

5    CESM results to the Greenland Ice sheet Mapping Project grid (1 x 1 km), which virtually removes this bias (green line in Figure 8), this implies that the CESM atmospheric model 'feels' a lower topography of the coastal GrIS than in reality, enhancing atmospheric and surface temperatures and rain in these elevations. However, since the model also produces too much snow at these elevations, we conclude that CESM tends to exaggerate the precipitation frequency of both snow and rain across the GrIS, rather than attributing the incorrect phase to precipitation. While acknowledging these model biases in

10    absolute precipitation frequencies, we argue that, overall, CESM reproduces the spatial patterns and seasonal cycle of snow and rain frequency satisfactorily well. This allows us to use CESM to analyze future changes in precipitation frequency on the GrIS.

### 3.3    Future changes in precipitation frequency

Next we use CESM with the radar simulator to analyze 21st century changes in the GrIS precipitation characteristics. To do so,

15    we compare the final 15-year period (2080-2095) of our simulation (referred to CC future) to our baseline CC present period (2006-2020).

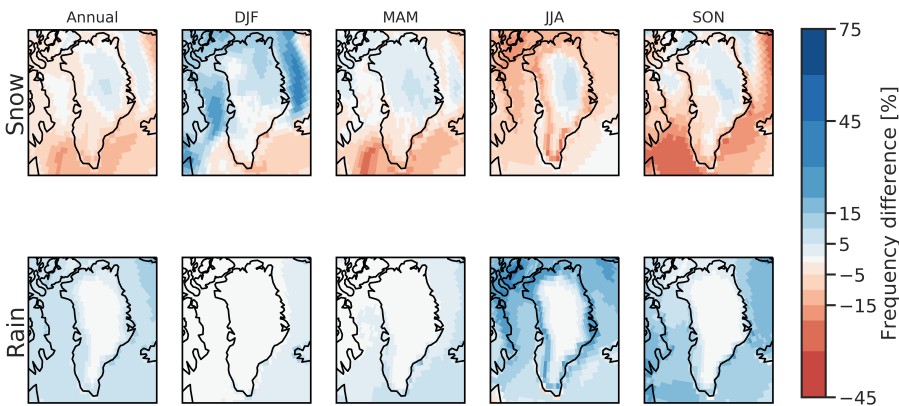

**Figure 9.** Annual (left) and seasonal (DJF, MAM, JJA, SON, from left to right) mean snowfall (top) and rainfall (bottom) frequency difference from present to future over the Greenland Ice Sheet simulated by CESM (present: 2006–2020 and future: 2080–2095).

The 21st century changes in precipitation frequency, as depicted in Figure 9, are substantial over the entire ice sheet. Across the south and much of the coast of the GrIS, annual snowfall frequency decreases by up to 10%. This contrasts the interior of the ice sheet, where annual snowfall frequency increases by up to 10%. This coastal decrease and interior increase is most clearly present in the summer (JJA), when coastal decreases in snow frequency exceed 20% to up to 40% in the southern GrIS. The increase in GrIS interior snow frequency is consistent throughout all seasons. The strongest increase in snow frequency occurs in winter, which is the season with the strongest simulated temperature increase in CESM (Peings et al., 2017). Snowfall and temperature are strongly correlated at low temperatures, since the Clausius-Clapeyron relationship dictates that the atmospheric saturation vapor pressure exponentially increases with temperature.Snowfall frequency over the oceanic regions surrounding the GrIS decreases throughout much of the year, although strong increases to the north are noted in winter, and to a lesser extent, in spring and fall. This snowfall increase is potentially associated with sea ice loss in these regions in the 21st century. More open water leads to enhanced atmospheric instability, condensation, and precipitation.

Rain frequency change shows a much more homogeneous signal across the GrIS and neighboring oceans (Figure 9). Annual rain frequency increases with 5-15% across the entirety of coastal GrIS, which essentially leads to a doubling of the present-day CESM rain frequency in these regions. While the winter season is still too cold for any rain on the GrIS at the end of the 21st century, rainfall occurs more frequently in spring, summer, and fall, and this frequency increase peaks in summer.

Averaged over the GrIS (Figure 10), the change in heavier snow frequency is slightly positive (0 to 2%) in winter, and negative in summer (down to -4% in August). Light snow frequency only changes substantially from June to October, with a decrease that also peaks in August (-4%). While heavier rain still doesn't occur on the GrIS at the end of the 21st century, light rain clearly increases, and dominates the change in snow frequency. At the end of the 21 century, light rain occurs in all months outside the core winter (November to March), which suggests that the rain-occurring season is extended with about 4 months relative to the present (Figure 6). In summer, the increase of light rain frequency peaks at almost 10%, which implies that rain frequency more than triples in summer relative to the present.

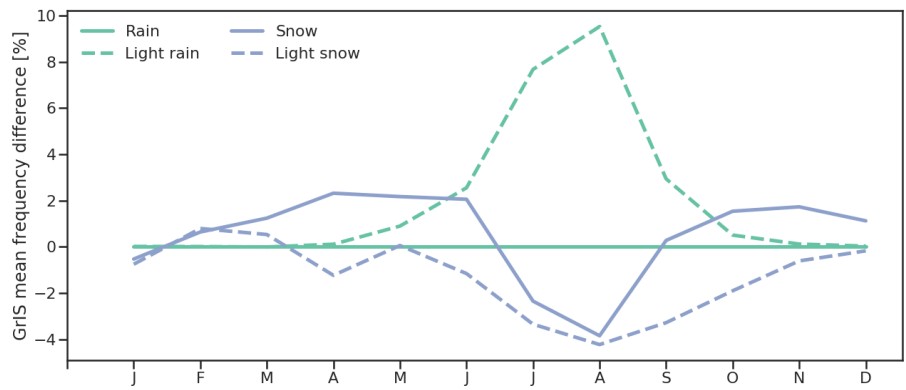

**Figure 10.** Greenland Ice Sheet averaged snowfall, light snowfall (blue: solid, dashed) and rainfall, light rainfall (green: solid, dashed) frequency difference between future (2080–2095) and present (2006–2020), as simulated by CESM.

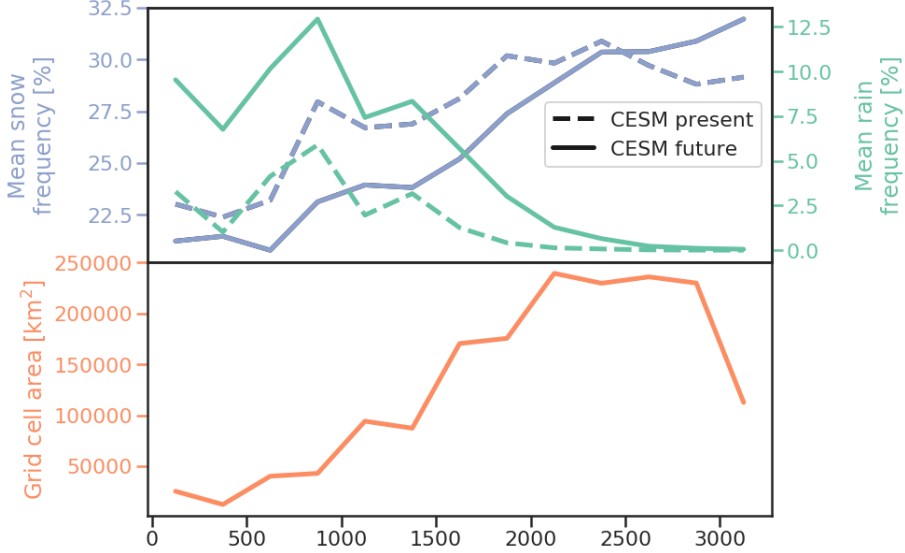

**Figure 11.** Top: Future (2080–2095) (solid) and present (2006–2020) (dashed) Greenland Ice Sheet snow (blue) and rain (green) frequency in 250-m elevation bins based on GIMP topography, as simulated by CESM. Bottom: total ice sheet surface area in each of these elevation bins according to GIMP.

The increase in (light) rain frequency is apparent over most of the GrIS (Figure 11), with roughly a tripling of rain frequency at all elevations below 2500 m a.s.l.. End of the 21st century rain frequency varies between 7 to 13% at elevations between 0 and 1500 m a.s.l., and decreases sharply above that elevation. However, the area of the GrIS that experiences at least some rain clearly extends inward and to higher elevations. Rain is projected to occur at elevations up to 2500 m a.s.l., in comparison to <2000 m a.s.l. in the present-day period. This exposes an additional area of >250,000 km$^2$ (>15%) of the GrIS to liquid

precipitation. In comparison to the rain changes, the changes in snow frequency are relatively small, with a small (0-2 %) decrease in snow frequency below 2500 m a.s.l., and a small increase (up to 2%) above that elevation, on the high GrIS interior. The relative minor change in snow frequency indicates that the increase in rain frequency is not completely compensated for a decrease in snow frequency. This finding signals that overall precipitation frequency is increasing over the GrIS, with an

increase of rain dominating over the entire ice sheet but the highest elevations, where rain does not occur and snow frequency increases.

## 4   Discussion and conclusions

In this paper, we used observations derived from active radar remote sensing (CloudSAT) and simulations with the Community Earth System Model to characterize precipitation frequency over the Greenland Ice Sheet. For the present-day climate, the

observations show that snowfall occurs frequently over the GrIS, with variations (1) in snowfall classification (light and heavier snow occur approximately equally frequently), (2) temporally throughout the year, and (3) spatially across the ice sheet. Rainfall, on the other hand, is rare, and only occurs in summer and at elevations below 2000 m a.s.l.. Our CloudSat results align well with previous studies. The snowfall frequency maximum of >30% over Southeast Greenland is consistent with various modeling results (Schuenemann et al., 2009; Hakuba et al., 2012; Berdahl et al., 2018, e.g., ). The summer maximum in

snowfall in the GrIS interior is confirmed by ground observations at Summit station (Castellani et al., 2015; Pettersen et al., 2018).

These observations were subsequently used to evaluate precipitation frequency output generated by CESM. The model is equipped with a satellite simulator, which allows for a consistent 'apples-to-apples' comparison with the observations. The results showed that, while CESM overestimates precipitation frequency on the GrIS overall, the model shows a realistic seasonal

cycle and spatial gradients. The differences between CESM and CloudSat are, at least partly, ascribed by the limited horizontal resolution (around 1 degree) of both products. Here we show that topography smoothing in CESM leads to underestimated precipitation frequency along the GrIS edges. While the native resolution of CloudSat is much higher (around 1 km), the aggregation of these observations in a 1 x 1 grid, along with ground clutter issues in steep topography, likely degrades the quality of this CloudSat product in accurately representing precipitation in the coastal regions of the GrIS (Bennartz et al., 2019).

To then analyze future changes in GrIS precipitation frequency, we analyzed CESM output for the end of the 21st century. The model suggests dramatic changes in the occurrence of rainfall, with rain occurrence extending in time (from April to October) and at much higher elevations (up to 2500 m a.s.l.). In contrast, snow frequency changes only marginally, and only increases across the high-elevation GrIS.

The comparison between CESM and CloudSat revealed clear biases in the simulated snow and rain frequency. This result

is consistent with the work of McIlhattan et al. (2017), who showed that the overestimated CESM snowfall frequency is potentially related to a exaggerated growth of cloud ice in expense of supercooled cloud liquid water in the model. The lack of supercooled liquid in polar clouds in CESM has been reported on previously (Miller et al., 2018; Kay et al., 2016a), and leads

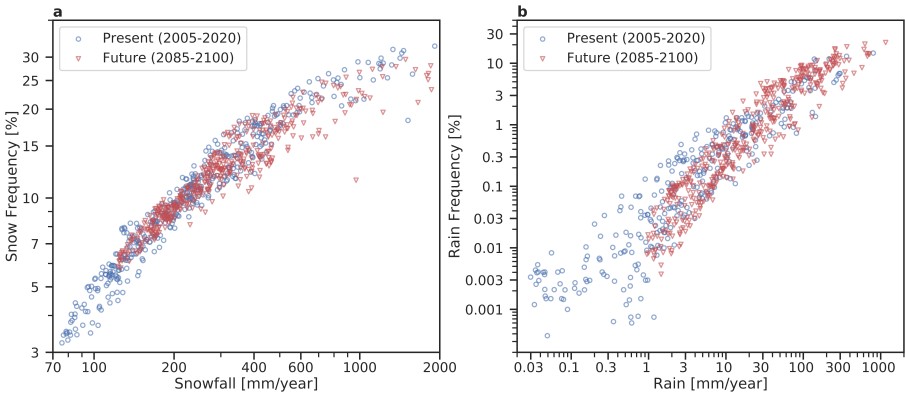

**Figure 12.** CESM simulated snowfall frequency (a) and rainfall frequency (light rain only, b) versus CESM simulated snowfall and rainfall rates in the present (2006–2020, blue circles) and future (2080–2095, red triangles).

to substantial biases in surface downwelling longwave radiation and surface temperature (Kay et al., 2016a), and GrIS surface melting.

Carefully recognizing these CESM biases, caution is warranted when quantitatively assessing simulated changes in the precipitation frequency throughout the 21st century. Doing so, we suggest to focus particularly on relative changes simulated
by CESM, which are likely more robust than the absolute changes. For example, while the absolute change in rainfall frequency is likely biased because the present-day rainfall frequency is overestimated, the simulated tripling of GrIS rainfall frequency is potentially a more robust change. In addition, this study only uses one climate model and one climate change scenario; to further test the robustness of our findings, future work should focus on using other models, with a satellite simulator embedded, and apply various climate change scenarios. Also, this study focuses on future changes in precipitation only, and does not give
a more general overview of future climate change on the GrIS that is provided by other studies using different versions of CESM (e.g., Vizcaíno et al., 2013; Muntjewerf et al., 2020).

Interpreting the relevance of these 21st century changes in precipitation frequency for the GrIS climate and mass balance, an outstanding question is how frequency relates to mass. For example, as rainfall frequency increases, does that imply that there is more mass of rain added to the GrIS surface? As we do not have reliable observations of precipitation fluxes from
CloudSat, we use CESM to analyze the relation between snow and rain frequency and the representative precipitation fluxes (Figure 12). Interestingly, for both snow and rain, the relation between precipitation frequency and rate apparent, with a near-linear increase in flux with frequency at low frequencies, and a much larger increase of flux with frequency as frequencies are higher. This relation, as suggested by CESM, indicates that even for small changes in precipitation frequency, precipitation rates change considerably; for example, an increase of snow frequency from 10 to 15% is associated with an approximate
doubling in snowfall rate (200 to 400 mm per year). That implies that a dramatic increase in rainfall frequency, as suggested by CESM, will be associated with much more rain on the GrIS. This has potential dramatic consequences for the GrIS surface conditions. In the GrIS ablation zone, slightly less snow in winter, and more rain in the transition seasons, will lead to more

rapid degradation of the winter snowpack, expediting exposure of bare ice in Spring and delaying ice burial in Fall. Rain falling on ice will decrease surface albedo, further enhancing melt, and the rain water will collect in surface lakes and streams that eventually end up in the ocean. In the percolation zone, less snow and much more rain will affect the storage capacity of the firn, and near-surface ice layer formation will lead to more rapid runoff of melt water. As CESM suggests an up- and inward

migration of the zone where rain occurs, a much larger area of the GrIS will be prone to summer runoff. Finally, the areas above 2500 m a.s.l., where our results indicate that snowfall increases slightly and rain still does not occur, will likely experience slight surface thickening.

In addition to understanding the impact of changing precipitation frequency on the GrIS surface, our methodology can be used to assess what a future CloudSat-style mission would observe in terms of changes in GrIS precipitation. Cloud radars

are, and will remain, essential to continually monitor polar precipitation, for a variety of reasons. Firstly, they measure at the right frequency: cloud radars (94 GHz such as CloudSat) provide the only spaceborne radar observations of high-latitude precipitation that have ever been made. Future missions currently will have this frequency: ESA's EarthCARE (https://earth.esa. int/web/guest/missions/esa-future-missions/earthcare; to be launched 2021) and NASA's ACCP mission (https://science.nasa. gov/earth-science/decadal-accp). In contrast, lower frequency precipitation radars (e.g., TRMM, GPM) cannot detect light

precipitation, which commonly occurs at high latitudes, including Greenland (as shown in this study). Secondly, CloudSat regularly samples the high latitude regions, whereas precipitation radars typically do not sample high latitude regions. Future cloud radar missions should continue to consider (near-)polar orbits to include high latitudes. Thirdly, co-locating spaceborne cloud radar with spaceborne lidar can help with assessment of light precipitation and precipitation phase. While our study only focused on CloudSat, future work should complement CloudSat radar retrievals with collocated CALIPSO lidar information to

study high latitude precipitation. Both the future EarthCARE and ACCP missions plan to include complimentary radar and lidar retrievals. Unfortunately, CloudSat only provides a 'curtain view' of cloud and precipitation vertical structures at high latitudes, and still provides relatively limited temporal coverage. Creative ways to combine CloudSat-like observations with meteorology can help isolate process-based relationships (e.g., Morrison et al., 2018; Gallagher et al., 2020). However, long-term (decadal or longer) data records are likely needed to isolate change from internal variability. For planning future Earth-observing missions,

satellite simulators can give a preliminary peek into potential findings, and provide initial assessments of how long a data record is needed to detect Greenland precipitation changes due to climate change. We suggest that future work leverages these tools, which has already been done for non-polar regions (e.g., Takahashi et al., 2019).

*Data availability.* CloudSat observations are available through the CloudSat Data Processing Center (https://cloudsat.atmos.colostate.edu/ data). The CESM data used in this study can be downloaded using Globus and this link: /glade/campaign/cesm/development/pcwg/jenkay/

b.e11.BRCP85C5CNBDRD.f09_g16.001_cosp1.4_opaq_prec_agl_precboth_opaq/. More information on using Globus on NCAR systems, please refer to https://www2.cisl.ucar.edu/resources/storage-and-file-systems/globus-file-transfers

*Competing interests.* The authors declare no competing interests.

*Acknowledgements.* The CESM project is supported primarily by the National Science Foundation. Computing and data storage resources, including the Cheyenne supercomputer (doi:10.5065/D6RX99HX), were provided by the Computational and Information Systems Laboratory (CISL) at NCAR. NCAR is sponsored by the National Science Foundation. This study is partially funded by NASA CloudSat, grant number 80NSSC20K0133.

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
