# Peer review of "Present-day and future Greenland Ice Sheet precipitation frequency from CloudSat observations and the Community Earth System Model"

_The Cryosphere, 2020_

## Referee Comment (RC1) · Anonymous Referee #1 · 2 Mar 2020

Lenaerts et al. use precipitation frequency observations from CloudSat to evaluate the accuracy of the Community Earth System Model (CESM) across the Greenland Ice Sheet. They find that CESM is able to reproduce present-day spatial patterns and seasonality of precipitation frequency fairly well. This lends confidence to their assessment of future precipitation frequency changes by 2080-2095. Overall, the paper is well-organized and easy to follow. The methods are clearly articulated and the results are thoroughly described. The discussion recognizes the limitations of CloudSat and the biases in CESM and presents a very insightful explanation about how we can use Earth System Models to make precipitation forecasts for the Greenland Ice Sheet. I have a few suggestions that would improve the manuscript which I detail below. The

main ones are to tidy up the first three paragraphs of the introduction and describe a bit more about how this study will actually aid future satellite planning campaigns. If the authors can address my relatively minor comments, I would be happy to endorse publication in The Cryosphere.

Title: I would urge the authors to consider replacing "satellite observations" with "Cloud-Sat" and "Earth System Model" with the "Community Earth System Model" in the title. Being more specific would probably make the paper more searchable.

P1 L16: "Clearly" is vague, how do we know that mass loss has accelerated? Models? GRACE?

P1 L19-20: Consider adding a reference that supports this statement.

P2 L3-4L: According to observations, precipitation decreased in western Greenland between 1996 and 2016 (Lewis et al. 2019; https://doi.org/10.5194/tc-13-2797-2019). Consider clarifying that this statement refers only to models.

P2 L7-9: There are two nice papers that were recently published in Science Advances that investigated this feedback. Consider referencing Noël et al. (2019; https://advances.sciencemag.org/content/5/9/eaaw0123) and Ryan et al. (2019; https://advances.sciencemag.org/content/5/3/eaav3738).

P3 L5-6: I was interested to learn about the "implications for future radar missions" but was disappointed that this was absent from the discussion and conclusions. Either remove this statement or discuss the implications for future radar missions in the manuscript. I would urge the latter to round off a very nice paper.

P3 L24: "heavy precipitation" do the authors mean "heavy rainfall"?

P8 L3-13: Might some of these uncertainties be explained by surface air temperatures in CESM being too warm/cold? If so, please discuss.

P9 L2-4: The authors miss an opportunity here to describe the future climate of Greenland according to an Earth System Model and how it differs from the present-day climate. What is the difference in mean average air temps? Is the seasonality of air temps weaker?

Figure 7b: Why would there be differences in the grid cell area between CloudSat and CESM? Please clarify.

—————————————————

---

## Referee Comment (RC2) · Anonymous Referee #2 · 30 Mar 2020

The manuscript presents a comparison between precipitation frequency as observed by CloudSat and as modeled by the Community Earth System Model for the last 15 years (2006-2020) and once determined that the model can simulate fairly well the pattern and the seasonal variability of precipitation, it extends the simulation to the last 15 years of the 21st Century. The results show a general decrease on snowfall frequency over the Greenland Ice Sheet except for the inner most elevated region where the snowfall frequency actually increases by 10%. Rainfall frequency is supposed to increase over the entire Greenland and will extend over higher elevation compared to present-days. These changes will affect climate and mass balance of the Greenland Ice Sheet with dramatic consequences on the snowpack.

The paper follows a pretty straightforward line of reasoning, clearly describing data, methodology and results. Here are few suggestions to clarify few points:

- The whole paper is about precipitation frequency so I suggest to explicitly describe what you mean with it (#precipitating pixels/#total pixels I suppose), especially because you also partition snow and rain and there could be confusion between the snow (rain) frequency over all pixels or over only precipitating pixels for example. - P4 L2: add a sentence to tell what RCP is (a forecast model? A future scenario? . . . ) - Section 3.1: describing present-day precipitation from CloudSat it is worth mentioning that there could be some biases due to the daylight operational mode CloudSat is operating on since 2011. The winter descending orbits over Southern Greenland for example do not provide any data. There are ongoing studies that will eventually quantify the biases but nothing has been published yet. I would suggest to mention the issue and just advice that no bias correction have been applied in the present study. - P5 L10-11: does the fact that areas below 2000m are actually a low percentage compared to other elevations affect precipitation frequencies? Describe a bit more how those variables are connected in your study. - P5 L19 and following: it is not really clear from your wording that you are calculating the difference between CESM and CloudSat frequencies. I would suggest to make this more explicit both here and on fig.5 caption. - Figure 3 caption: the "grid-cell" area concept is not really clear, is it the total area of the GrIS for each elevation bin? I feel like this "grid-cell" is a bit misleading. - P9 L1-4: as a curiosity, have you tried to compare different intervals like only 10 years or for example 15 years around 2050? Just to see if there is a dependence on the actual interval of years used for the comparison or if we can extend the results independently on that. - In general I feel like the conclusion section is a bit weak, some more information could be added about the mass balance and what to expect for future satellite missions here. Better describing the consequences of your results would absolutely help making the paper stronger.

Minor comments:
- In the abstract you mention RCP8.5, I suggest to explicitly write "Representative Concentration Pathway 8.5 Scenario" as you did for CESM earlier. - P2 L8: why the snow layer "needs" to be melted away? - P2 L29: remove the first "the" from "CloudSat is the currently the. . ." - P2 L30: there aren't so many papers about the observational assessment of Antarctic Ice Sheet precipitation rates, so I would mention all of them, including Milani et al 2018 that considered also the surrounding oceans and the ground clutter corrections. - P3 L21: "gridded observations", can you give some details about the grid you used for this work? - P3 L21: "2CPC", for a reader not familiar with CloudSat this acronym does not make much sense, explicitly mention 2C-PRECIPITATION-COLUMN product so they know what to look for. - P6 L4: you probably forgot to say "in winter". - P8 L5: Is Figure A1 an appendix figure? It is not clear at the end of the manuscript that there is an appendix, shouldn't this figure just be included as a regular figure since there is no in-depth description in a real supplemental section? - P8 L7: what is GIMP and what is its grid? The full name is provided in fig. 3 caption, I would suggest to include it in the text as well. - P9 L14-21: I guess you are referring to figure 9, please cite it. - P10 L2: "the increase of light frequency", add "light rain frequency".

---

## Referee Comment (RC3) · Anonymous Referee #3 · 31 Mar 2020

Review of tc-2020-31

This manuscript examines the current state of precipitation around the Greenland Ice Sheet (GrIS) using precipitation products from CloudSat (CS). It partitions the frequency of the snowfall into regions of the GrIS as well as looks at impacts based on elevation. The manuscript looks at both snow (moderate and light) and rain (light only) and compares to both CESM findings of current day and future projections. In general, CESM overestimates the rainfall frequency, but reproduces the spatial and seasonal variability when compared to CS. Under future warming conditions, the authors find that rainfall will increase at higher elevations of the GrIS, whereas snow only increases

in the highest elevations (>2500 MASL).

Overall, this is a well-written and organized manuscript. I believe that the material is novel and will add to our understanding of future impacts of precipitation to the GrIS. I have only some minor comments and suggestions of added references in some areas where I think they would help broaden or support the manuscript. These are listed below:

* The introduction could benefit with a little more background and citations (especially the first three paragraphs). For example, please cite: ..."equivalent to 7.3 meter sea 15 level equivalent." (P1, L15), ..."driven by a progressively declining SMB." (P1, L20). Also, could you add any comments on recent data from GRACE or IceSat2/IceBridge in constraining some of these measures of SMB somewhere in the Intro?

* Could you please add McIlhattan et al. (2019 TCD – in revisions) as well at "and GrIS precipitation rates (Bennartz et al., 2019)" (P2, L31)? McIlhattan et al. also examines the frequency and rates of snowfall over the GrIS (https://www.the-cryosphere-discuss.net/tc-2019-223/tc-2019-223.pdf)

* This comment relates to what you say on P2, L33: "In particular, CloudSat radar reflectivity profiles are contaminated by ground clutter in the bottom kilometer of the atmosphere..." Both Bennartz et al. (2019) and McIlhattan et al. (2019) examine the impact of the ground clutter and the accuracy of the lowest available bin on snow rate information. McIlhattan et al. found that up to 25% of the light snow-producing mixed-phase clouds are likely being missed by CS, when compared to studies at Summit Station (Pettersen et al., 2018 (ACP)). It might be good to have some discussion of this in the data and methods section. I do not think that it fits in the introduction and I do not think it will detract from the overall narrative, but I think some discussion or inclusion of the ground clutter/detection issues in the Data and Methods section would be helpful. It might also be helpful to show the definitions of "light" versus "regular" snow and rain in the methods (I did find it in Kay et al., 2018, but it would be nice to include here as

well).

* P4, L9-11 I would add some citations of previous precipitation studies that agree with these findings. For example, "to >30% over Southeast Greenland" is consistent with previous studies, such as: Schuenemann et al., 2009; Hakuba et al., 2012; Berdahl et al., 2018. And "The interior experiences snowfall most frequently in the summer (JJA, >20%)," is in line with ground-based studies from Summit Station, so I suggest noting that and adding the citations: Castellani et al., 2015 (https://agupubs.onlinelibrary.wiley.com/doi/full/10.1002/2015JD023072) and Pettersen et al., 2018 (https://www.atmos-chem-phys.net/18/4715/2018/acp-18-4715-2018.pdf). Throughout this paragraph, it would be helpful to note previous work that is consistent with these findings (similar with the rain frequencies).

* Figure 3 caption implies there should be dashed lines, but they are not shown. It does say "not shown" in the text (P5, L7). I think it would be nice to show these. Also, this is in agreement with what McIllhattan et al. found (see figure 7).

* P6, L1: You say "In contrast, interior GrIS summer snowfall frequency is slightly lower in CESM than in CloudSat." Both Pettersen et al. (2018) and McIlhattan et al. found that mixed-phase clouds were the dominate cloud type producing snowfall in the summer (as opposed to deep, frontal clouds). CS misses many of these lightly precipitating mixed-phase clouds (especially over the interior where CS was compared to Summit Station instrumentation). Is it worth noting this point? Either here or in the discussion? It could be that CS is missing some of this summertime precipitation that is actually being modeling correctly?

* Figure 6: just a comment that not only does the heavier snow seem to have less of a seasonal cycle, it seems to be completely missing the uptick in SON that is due to the firing up of the NA storm track. Just a comment – but does CESM not accurately capture the NA storm tracks impinging the GrIS?

* P8, L3-4: "A part of these discrepancies between CESM and CloudSat may be ascribed to CESM (at its horizontal resolution of 1 degree) not resolving the steep topography and related surface climate and precipitation gradients of the marginal GrIS" – also, Bennartz et al., (2019) showed that CS additionally has a very difficult time resolving the precipitation accurately in the steep topographic regions (as well as other studies focused on CS and GPM orographic impacts). Could it also be that both CESM and CS have difficulties here? Might be worth noting – I am not sure I would say it is all CESM.

\* P9, L9 -10: "The increase in GrIS interior snow frequency is consistent throughout all seasons, and most prominent in winter (DJF)" – any speculation as to why? Is it temperature driven, moisture? (either here or in the discussion).

\* Figure A1 is not really in an appendix. Is it worth just adding it as a regular figure? Or adding a proper Appendix with some verbiage?

\* Final comment – Much of the above comments/citations could be added either where I noted or in the discussion. I think adding some of the above gives the paper more context.

---

## Author Comment (AC1) · 2 Jun 2020

**Reviewer #1**

Lenaerts et al. use precipitation frequency observations from CloudSat to evaluate the accuracy of the Community Earth System Model (CESM) across the Greenland Ice Sheet. They find that CESM is able to reproduce present-day spatial patterns and seasonality of precipitation frequency fairly well. This lends confidence to their assessment of future precipitation frequency changes by 2080-2095. Overall, the paper is well-organized and easy to follow. The methods are clearly articulated and the results are thoroughly described. The discussion recognizes the limitations of CloudSat and the biases in CESM and presents a very insightful explanation about how we can use Earth System Models to make precipitation forecasts for the Greenland Ice Sheet. I have a few suggestions that would improve the manuscript which I detail below. The main ones are to tidy up the first three paragraphs of the introduction and describe a bit more about how this study will actually aid future satellite planning campaigns. If the authors can address my relatively minor comments, I would be happy to endorse publication in The Cryosphere.

Title: I would urge the authors to consider replacing "satellite observations" with "CloudSat" and "Earth System Model" with the "Community Earth System Model" in the title. Being more specific would probably make the paper more searchable.

We changed the title accordingly.

P1 L16: "Clearly" is vague, how do we know that mass loss has accelerated? Models? GRACE?

We changed this sentence to 'observations indicate that GrIS mass loss has accelerated'

P1 L19-20: Consider adding a reference that supports this statement.

We added a new reference to support this statement: the new IMBIE assessment of Greenland Ice Sheet mass balance (Shepherd et al., 2020)

Shepherd, A., Ivins, E., Rignot, E., Smith, B., van den Broeke, M., Velicogna, I., et al. (2020). Mass balance of the Greenland Ice Sheet from 1992 to 2018. *Nature*, *579*(7798), 233–239.

P2 L3-4L: According to observations, precipitation decreased in western Greenland between 1996 and 2016 (Lewis et al. 2019; https://doi.org/10.5194/tc-13-2797-2019). Consider clarifying that this statement refers only to models.

Thanks, we added 'climate modeling indicates that' and added the Lewis et al., 2019 to the statement 'with only an increase over parts of the interior'.

P2 L7-9: There are two nice papers that were recently published in Science Ad- vances that investigated this feedback. Consider referencing Noël et al. (2019; https://advances.sciencemag.org/content/5/9/eaaw0123) and Ryan et al. (2019; https://advances.sciencemag.org/content/5/3/eaav3738).

Both references are added to the revised manuscript.

P3 L5-6: I was interested to learn about the "implications for future radar missions" but was disappointed that this was absent from the discussion and conclusions. Either remove this statement or discuss the implications for future radar missions in the manuscript. I would urge the latter to round off a very nice paper.

We agree with the reviewer (as well as the other reviewers) that a discussion of this is a welcome addition to our paper and increases its significance. Therefore we added this paragraph to the discussion:

"In addition to understanding the impact of changing precipitation frequency on the GrIS surface, our methodology can be used to assess what a future CloudSat-style mission would observe in terms of changes in GrIS precipitation. Cloud radars are, and will remain, essential to continually monitor polar precipitation, for a variety of reasons. Firstly, they measure at the right frequency: cloud radars (94 GHz such as CloudSat) provide the only spaceborne radar observations of high-latitude precipitation that have ever been made. Future missions currently will have this frequency: ESA's EarthCARE (https://earth.esa.int/web/guest/missions/esa-future-missions/earthcare ; to be launched 2021) and NASA's ACCP mission (https://science.nasa.gov/earth-science/decadal-accp). In contrast, lower frequency precipitation radars (e.g., TRMM, GPM) cannot detect light precipitation, which commonly occurs at high latitudes, including Greenland (as shown in this study). Secondly, CloudSat regularly samples the high latitude regions, whereas precipitation radars typically do not sample high latitude regions. Future cloud radar missions should continue to consider (near-)polar orbits to include high latitudes. Thirdly, co-locating spaceborne cloud radar with spaceborne lidar can help with assessment of light precipitation and precipitation phase. While our study only focused on CloudSat, future

work should complement CloudSat radar retrievals with collocated CALIPSO lidar information to study high latitude precipitation. Both the future EarthCARE and ACCP missions plan to include complimentary radar and lidar retrievals.

Unfortunately, CloudSat only provides a 'curtain view' of cloud and precipitation vertical structures at high latitudes, and still provides relatively limited temporal coverage. Creative ways to combine CloudSat-like observations with meteorology can help isolate process-based relationships (e.g., Morrison et al. 2018; Gallagher et al. 2020). However, long-term (decadal or longer) data records are likely needed to isolate change from internal variability. For planning future Earth-observing missions, satellite simulators can give a preliminary peek into potential findings, and provide initial assessments of how long a data record is needed to detect Greenland precipitation changes due to climate change. We suggest that future work leverages these tools, which has already been done for non-polar regions (e.g., Takahashi et al. 2019)."

P3 L24: "heavy precipitation" do the authors mean "heavy rainfall"?

We refer both to snow and rain here, so precipitation seems to be valid to use.

P8 L3-13: Might some of these uncertainties be explained by surface air temperatures in CESM being too warm/cold? If so, please discuss.

We would argue that, although surface air temperatures partly determine the precipitation phase (and frequency as well), many factors control precipitation formation (e.g. cloud microphysics, thermodynamics of full atmospheric column, advection, etc.). We think that is clearly beyond the scope of this study to perform a detailed evaluation of CESM1 precipitation formation mechanisms

P9 L2-4: The authors miss an opportunity here to describe the future climate of Greenland according to an Earth System Model and how it differs from the present-day cli- mate. What is the difference in mean average air temps? Is the seasonality of air temps weaker?

The goal of this study is not to provide a general overview of the future climate on the Greenland ice sheet, but to focus on precipitation phase and frequency chases in particular. Other studies have focused on using various versions of CESM to characterize future changes in Greenland climate, e.g. Vizcaino et al., 2014; Muntjewerf et al., 2020. These references are added to the discussion: "Also, this study focuses on future changes in precipitation only, and does not give a more general overview of

future climate change on the GrIS that is provided by other studies using different versions of CESM (e.g. Vizcaino et al., 2013; Muntjewerf et al., 2020)."

Vizcaíno, M., Lipscomb, W. H., Sacks, W. J., & van den Broeke, M. (2013). Greenland Surface Mass Balance as Simulated by the Community Earth System Model. Part II: Twenty-First-Century Changes. Journal of Climate, 27(1), 215–226. https://doi.org/10.1175/JCLI-D-12-00588.1

Muntjewerf, L., Petrini, M., Vizcaino, M., Ernani da Silva, C., Sellevold, R., Scherrenberg, M. D. W., et al. (2020). Greenland Ice Sheet Contribution to 21st Century Sea Level Rise as Simulated by the Coupled CESM2.1-CISM2.1. Geophysical Research Letters, 47(9), e2019GL086836. https://doi.org/10.1029/2019GL086836

Figure 7b: Why would there be differences in the grid cell area between CloudSat and CESM? Please clarify.

We acknowledge that 'grid cell area' is confusing; this is simply the total area in each bin. We have changed this to 'Area' instead and divided the number by $10^5$ to improve readability of the labels.

---

## Author Comment (AC2) · 2 Jun 2020

**Reviewer #2**

The manuscript presents a comparison between precipitation frequency as observed by CloudSat and as modeled by the Community Earth System Model for the last 15 years (2006-2020) and once determined that the model can simulate fairly well the pattern and the seasonal variability of precipitation, it extends the simulation to the last 15 years of the 21st Century. The results show a general decrease on snowfall frequency over the Greenland Ice Sheet except for the inner most elevated region where the snowfall frequency actually increases by 10%. Rainfall frequency is supposed to increase over the entire Greenland and will extend over higher elevation compared to present-days. These changes will affect climate and mass balance of the Greenland Ice Sheet with dramatic consequences on the snowpack. The paper follows a pretty straightforward line of reasoning, clearly describing data, methodology and results. Here are few suggestions to clarify few points:

We thank the reviewer for their positive feedback. We provide a response to all items below.

The whole paper is about precipitation frequency so I suggest to explicitly describe what you mean with it (#precipitating pixels/#total pixels I suppose), especially because you also partition snow and rain and there could be confusion between the snow (rain) frequency over all pixels or over only precipitating pixels for example.

Good point, we added to the first paragraph of the data and methods section: "We define precipitation frequency as the ratio between the number of time steps with precipitation and the total number of time steps. If averaged across an area, such as the ice sheet or elevation bin, frequency is defined as the average frequency of all grid cells contained within that area."

P4 L2: add a sentence to tell what RCP is (a forecast model? A future scenario? . . . )

We added: "using the worst-case Representative Concentration Pathway (RCP) 8.5 greenhouse gas emissions scenario."

Section 3.1: describing present-day precipitation from CloudSat it is worth mentioning that there could be some biases due to the daylight operational mode CloudSat is operating on since 2011. The winter descending orbits over Southern Greenland for example do not provide any data. There are ongoing studies that will eventually quantify

the biases but nothing has been published yet. I would suggest to mention the issue and just advice that no bias correction have been applied in the present study.

Thanks for pointing that out. We have added this to the Data and methods section:" Since CloudSat has been operating on daytime only mode since 2011, which might potentially introduce biases that are not considered in this study."

P5 L10-11: does the fact that areas below 2000m are actually a low percentage compared to other eleva- tions affect precipitation frequencies? Describe a bit more how those variables are connected in your study.

Yes, it does, and this sentence aims to address exactly that: "This implies that, although all areas below 2000 m a.s.l. experience rain, all these elevation bands combined only occupy ≈ 38% of the ice sheet area."

P5 L19 and following: it is not really clear from your wording that you are calculating the difference between CESM and CloudSat frequencies. I would suggest to make this more explicit both here and on fig.5 caption.

Good point, we added a sentence to start the paragraph: "We first present the CESM precipitation frequencies (Figure 4), and then compare them directly to CloudSat (Figure 5)." In the caption of Figure 5, we more explicitly state that these map shows CESM-CloudSat: "Present-day, annual (left) and seasonal (DJF, MAM, JJA, SON, from left to right) mean snowfall (top) and rainfall (bottom) frequency difference between CESM (2006--2020) and CloudSat 2CPC (2006--2016). Positive values indicate that CESM overestimates precipitation frequency relative to CloudSat."

Figure 3 caption: the "grid-cell" area concept is not really clear, is it the total area of the GrIS for each elevation bin? I feel like this "grid-cell" is a bit misleading.

We acknowledge that 'grid cell area' is confusing; this is simply the total area in each bin. We have changed this to 'Area' instead and divided the number by 10^5 to improve readability of the labels.

P9 L1-4: as a curiosity, have you tried to compare different intervals like only 10 years or for example 15 years around 2050? Just to see if there is a dependence on the actual interval of years used for the comparison or if we can extend the results independently on that.

We have not done that. The 15 years is chosen because it is a compromise between having sufficient number of years to filter out much of the internal variability, and to stay as close to the end of the 21st century to see the largest signal.

In general I feel like the conclusion section is a bit weak, some more information could be added about the mass balance and what to expect for future satellite missions here. Better describing the consequences of your results would absolutely help making the paper stronger.

We agree, and have added new text to discuss the relevance for future satellite missions.

Minor comments:

In the abstract you mention RCP8.5, I suggest to explicitly write "Representative Concentration Pathway 8.5 Scenario" as you did for CESM earlier.

Done.

P2 L8: why the snow layer "needs" to be melted away?

Changed to 'is melted away'

P2 L29: remove the first "the" from "CloudSat is the currently the. . ."

Done.

P2 L30: there aren't so many papers about the observational assess- ment of Antarctic Ice Sheet precipitation rates, so I would mention all of them, including Milani et al 2018 that considered also the surrounding oceans and the ground clutter corrections.

Thanks, added Milani et al., 2018 and Lemonnier et al., 2020 references to revised paper.

Milani, L., Kulie, M. S., Casella, D., Dietrich, S., L'Ecuyer, T. S., Panegrossi, G., et al. (2018). CloudSat snowfall estimates over Antarctica and the Southern Ocean: An assessment of independent retrieval methodologies and multi-year snowfall analysis. *Atmospheric Research*, *213*, 121–135. https://doi.org/10.1016/j.atmosres.2018.05.015

Lemonnier, F., Madeleine, J.-B., Claud, C., Palerme, C., Genthon, C., L'Ecuyer, T., & Wood, N. B. (2020). CloudSat-Inferred Vertical Structure of Snowfall Over the Antarctic Continent. *Journal of Geophysical Research: Atmospheres*, *125*(2), e2019JD031399. https://doi.org/10.1029/2019JD031399

P3 L21: "gridded observations", can you give some details about the grid you used for this work?

This grid is a 1x1 degree grid in which all CloudSat 2C-PRECIPITATION-COLUMN are aggregated. This is added to the text.

P3 L21: "2CPC", for a reader not familiar with CloudSat this acronym does not make much sense, explicitly mention 2C-PRECIPITATION-COLUMN product so they know what to look for.

Done.

P6 L4: you probably forgot to say "in winter".

We are unclear as to what the reviewer refers to, since the sentence includes 'in winter'.

P8 L5: Is Figure A1 an appendix figure? It is not clear at the end of the manuscript that there is an appendix, shouldn't this figure just be included as a regular figure since there is no in-depth description in a real supplemental section?

We added this figure as a separate regular figure in the revised manuscript.
P8 L7: what is GIMP and what is its grid? The full name is provided in fig. 3 caption, I would suggest to include it in the text as well.

Done.

P9 L14-21: I guess you are referring to figure 9, please cite it.

Done.

 P10 L2: "the increase of light frequency", add "light rain frequency".

Done.

---

## Author Comment (AC3) · 2 Jun 2020

**Reviewer #3**

Review of tc-2020-31

This manuscript examines the current state of precipitation around the Greenland Ice Sheet (GrIS) using precipitation products from CloudSat (CS). It partitions the frequency of the snowfall into regions of the GrIS as well as looks at impacts based on elevation. The manuscript looks at both snow (moderate and light) and rain (light only) and compares to both CESM findings of current day and future projections. In general, CESM overestimates the rainfall frequency, but reproduces the spatial and seasonal variability when compared to CS. Under future warming conditions, the authors find that rainfall will increase at higher elevations of the GrIS, whereas snow only increases in the highest elevations (>2500 MASL).

Overall, this is a well-written and organized manuscript. I believe that the material is novel and will add to our understanding of future impacts of precipitation to the GrIS. I have only some minor comments and suggestions of added references in some areas where I think they would help broaden or support the manuscript. These are listed below:

We thank the reviewer for their positive assessment. We provide a point-by-point response below.

* The introduction could benefit with a little more background and citations (especially the first three paragraphs). For example, please cite: . . ."equivalent to 7.3 meter sea 15 level equivalent." (P1, L15), . . ."driven by a progressively declining SMB." (P1, L20). Also, could you add any comments on recent data from GRACE or IceSat2/IceBridge in constraining some of these measures of SMB somewhere in the Intro?

Thanks for this suggestion. We have added several new references to the intro.

Morlighem et al., 2017 - to support the 7.4 m sea level rise equivalent claim.

Shepherd et al., 2019: to support 'progressively declining SMB'

Montgomery et al., 2020: Constraining SMB using IceBridge in SW Greenland

Fettweis et al., 2020: model intercomparison of Greenland SMB

Morlighem, M., Williams, C. N., Rignot, E., An, L., Arndt, J. E., Bamber, J. L., et al. (2017). BedMachine v3: Complete Bed Topography and Ocean Bathymetry Mapping of Greenland From Multibeam Echo Sounding Combined With Mass Conservation.

*Geophysical Research Letters*, 44(21), 11,051-11,061.
https://doi.org/10.1002/2017GL074954

Montgomery, L., Koenig, L., Lenaerts, J. T. M., & Kuipers Munneke, P. (2020).
Accumulation rates (2009-2017) in Southeast Greenland derived from airborne snow
radar and comparison with regional climate models. *Annals of Glaciology*, 1–9.
https://doi.org/10.1017/aog.2020.8

Fettweis, X., Hofer, S., Krebs-Kanzow, U., Amory, C., Aoki, T., Berends, C. J., et al.
(2020). GrSMBMIP: Intercomparison of the modelled 1980--2012 surface mass balance
over the Greenland Ice sheet. *The Cryosphere Discussions*, *2020*, 1–35.
https://doi.org/10.5194/tc-2019-321

Shepherd, A., Ivins, E., Rignot, E., Smith, B., van den Broeke, M., Velicogna, I., et al.
(2020). Mass balance of the Greenland Ice Sheet from 1992 to 2018. *Nature*,
*579*(7798), 233–239. https://doi.org/10.1038/s41586-019-1855-2

* Could you please add McIlhattan et al. (2019 TCD – in revisions) as well at "and GrIS
precipitation rates (Bennartz et al., 2019)" (P2, L31)? McIlhattan et al. also ex- amines
the frequency and rates of snowfall over the GrIS (https://www.the-cryosphere-
discuss.net/tc-2019-223/tc-2019-223.pdf)

Done.

* This comment relates to what you say on P2, L33: "In particular, CloudSat radar
reflectivity profiles are contaminated by ground clutter in the bottom kilometer of the
atmosphere. . ." Both Bennartz et al. (2019) and McIlhattan et al. (2019) examine the
impact of the ground clutter and the accuracy of the lowest available bin on snow rate
information. McIlhattan et al. found that up to 25% of the light snow-producing mixed-
phase clouds are likely being missed by CS, when compared to studies at Summit
Station (Pettersen et al., 2018 (ACP)). It might be good to have some discussion of this
in the data and methods section. I do not think that it fits in the introduction and I do not
think it will detract from the overall narrative, but I think some discussion or inclusion of
the ground clutter/detection issues in the Data and Methods section would be helpful. It
might also be helpful to show the definitions of "light" versus "regular" snow and rain in
the methods (I did find it in Kay et al., 2018, but it would be nice to include here as well).

Thanks for pointing that out. We added to the Data and methods, after the first sentence
"In addition, CloudSat suffers from ground clutter, which leads to, for example, missing

up to 25% of the light snow producing mixed-phase clouds over central Greenland (Bennartz et al., 2019; McIlhattan et al., 2019).”

We also added the thresholds to the text in the Data and methods.

* P4, L9-11 I would add some citations of previous precipitation studies that agree with these findings. For example, “to >30% over Southeast Greenland” is con- sistent with previous studies, such as: Schuenemann et al., 2009; Hakuba et al., 2012; Berdahl et al., 2018. And “The interior experiences snowfall most fre- quently in the summer (JJA, >20%),” is in line with ground-based studies from Summit Station, so I suggest noting that and adding the citations: Castellani et al., 2015 (https://agupubs.onlinelibrary.wiley.com/doi/full/10.1002/2015JD023072) and Pettersen et al., 2018 (https://www.atmos-chem-phys.net/18/4715/2018/acp-18-4715- 2018.pdf). Throughout this paragraph, it would be helpful to note previous work that is consistent with these findings (similar with the rain frequencies).

Thanks once again for providing these references, these are useful to put our results into a perspective. To clearly separate our own results and the discussion, we have added a brief discussion on the comparison/agreement with existing studies to the discussion (first paragraph), adding the references suggested by the reviewer: ”Our CloudSat results align well with previous studies. The snowfall frequency maximum of >30% over Southeast Greenland is consistent with various modeling results ( Schuenemann et al., 2009; Hakuba et al., 2012; Berdahl et al., 2018). The summer maximum in snowfall in the GrIS interior is confirmed by ground observations at Summit station (Castellani et al., 2015; Pettersen et al, 2018).”

* Figure 3 caption implies there should be dashed lines, but they are not shown. It does say “not shown” in the text (P5, L7). I think it would be nice to show these. Also, this is in agreement with what McIllhattan et al. found (see figure 7).

We revised Figure 3 accordingly.

* P6, L1: You say “In contrast, interior GrIS summer snowfall frequency is slightly lower in CESM than in CloudSat.” Both Pettersen et al. (2018) and McIlhattan et al. found that mixed-phase clouds were the dominate cloud type producing snowfall in the summer (as opposed to deep, frontal clouds). CS misses many of these lightly precipitating mixed-phase clouds (especially over the interior where CS was compared to Summit Station instrumentation). Is it worth noting this point? Either here or in the discussion? It could be that CS is missing some of this summertime precipitation that is actually being modeling correctly?

That is a fair point, but if we understand correctly, this would actually aggravate the CESM bias, since the model produces even less light snowfall than CloudSat (which - as the reviewer indicates - misses a fraction of these events). Since we already mentioned that CloudSat potentially fails to detect such events in the Data and methods section, we would argue that this topic has been addressed sufficiently.

* Figure 6: just a comment that not only does the heavier snow seem to have less of a seasonal cycle, it seems to be completely missing the uptick in SON that is due to the firing up of the NA storm track. Just a comment – but does CESM not accurately capture the NA storm tracks impinging the GrIS?

We are not aware of a study that analyzes the CESM1 storm track seasonality in/around the North Atlantic region, but this is potentially the case. Since this topic is out of the scope of our study, and would require substantial additional analysis, we prefer to refrain from mentioning it.

* P8, L3-4: "A part of these discrepancies between CESM and CloudSat may be ascribed to CESM (at its horizontal resolution of 1 degree) not resolving the steep topog- raphy and related surface climate and precipitation gradients of the marginal GrIS" – also, Bennartz et al., (2019) showed that CS additionally has a very difficult time re- solving the precipitation accurately in the steep topographic regions (as well as other studies focused on CS and GPM orographic impacts). Could it also be that both CESM and CS have difficulties here? Might be worth noting – I am not sure I would say it is all CESM.

Good point. We added to the discussion: "The differences between CESM and CloudSat are, at least partly, ascribed by the limited horizontal resolution (around 1 degree) of both products. Here we show that topography smoothing in CESM leads to underestimated precipitation frequency along the GrIS edges, but we should note that CloudSat also struggles to accurately represent precipitation in steep topographic regions (Bennartz et al., 2019)."

* P9, L9 -10: "The increase in GrIS interior snow frequency is consistent throughout all seasons, and most prominent in winter (DJF)" – any speculation as to why? Is it temperature driven, moisture? (either here or in the discussion).

We added in the discussion (second paragraph): "The strongest increase in snow frequency occurs in winter, which is the season with the strongest simulated temperature increase in CESM (Peings et al., 2017). Snowfall and temperature are strongly correlated at low temperatures, since the Clausius-Clapeyron relationship

dictates that the atmospheric saturation vapor pressure exponentially increases with temperature."

Peings, Y., Cattiaux, J., Vavrus, S., & Magnusdottir, G. (2017). Late Twenty-First-Century Changes in the Midlatitude Atmospheric Circulation in the CESM Large Ensemble. *Journal of Climate*, *30*(15), 5943–5960. https://doi.org/10.1175/JCLI-D-16-0340.1

* Figure A1 is not really in an appendix. Is it worth just adding it as a regular figure? Or adding a proper Appendix with some verbiage?

We have added this figure as a regular figure.

* Final comment – Much of the above comments/citations could be added either where I noted or in the discussion. I think adding some of the above gives the paper more context.